# Multi-modal Dependency Tree for Video Captioning

**Wentian Zhao, Xinxiao Wu**[*]
Beijing Laboratory of Intelligent Information Technology
School of Computer Science
Beijing Institute of Technology
No. 5, Zhongguancun South Street, Beijing, China
{wentian_zhao,wuxinxiao}@bit.edu.cn

**Jiebo Luo**
Department of Computer Science
University of Rochester
Rochester, NY 14627 USA
jluo@cs.rochester.edu

## Abstract

Generating fluent and relevant language to describe visual content is critical for the video captioning task. Many existing methods generate captions using sequence models that predict words in a left-to-right order. In this paper, we investigate a graph structured model by explicitly modeling the hierarchical structure in the sentences to further improve the fluency and relevance of the generated captions. To this end, we propose a novel video captioning method that generates a sentence by first constructing a multi-modal dependency tree and then traversing the constructed tree, where the syntactic structure and semantic relationship in the sentence are represented by the tree topology. To take full advantage of the information from both vision and language, both the visual and textual representation features are encoded into each tree node. Different from existing dependency parsing methods that generate uni-modal dependency trees for language understanding, our method constructs multi-modal dependency trees for language generation of videos. We also propose a tree-structured reinforcement learning algorithm to effectively optimize the captioning model, where a novel reward is designed by evaluating the semantic consistency between the generated sub-trees and the ground-truth tree. Extensive experiments on several video captioning datasets demonstrate the effectiveness of the proposed method.

## 1 Introduction

Video captioning has received extensive attention from researchers in both computer vision and natural language processing. It is a challenging task since it requires not only understanding the visual content but also generating fluent and relevant sentences. With the success of deep neural networks in natural language processing, many existing video captioning methods [1, 2] use recurrent neural network (RNN) to generate sentences, which processes sequences by updating hidden states. Several recent studies [3–6] apply Transformer [7], which relates the words in the sequences using a self-attention mechanism, to video captioning. All these methods treat each sentence as a word sequence and generate words in a predefined left-to-right order by capturing the relatively close contextual relationship between words in the sentence.

In this paper, we investigate a *graph structured model* for video captioning to explicitly model the *hierarchical structure in the sentence* and capture the *long-range dependency between words*. With this in mind, we propose to generate a sentence by first constructing a multi-modal dependency tree in a top-down and depth-first order, and then traversing the tree in a recursive manner, as shown in Figure 1. Each node of the tree is represented by a multi-modal embedding representation that integrates the information from both visual and textual modalities. During the tree construction, each

---
[*]Corresponding author.

35th Conference on Neural Information Processing Systems (NeurIPS 2021).

newly generated node receives the multi-modal embeddings from its parent and sibling nodes, and these multi-modal embeddings are used to predict the attention weights of the input features and the word. The attended visual feature and the predicted word are fused to generate the multi-modal embedding of the new node.

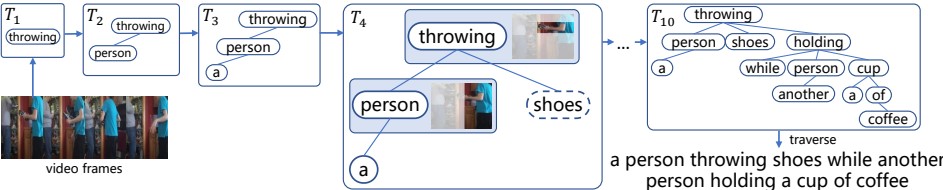

Figure 1: The sentence generation process of the proposed method. $T_i$ denotes the dependency tree generated in the $i$-th step, and the colored box denotes the multi-modal embedding of each node.

Different from existing sequence models that tend to focus on the dependency between each word and its close preceding words, our tree model sufficiently captures the global dependency structure in the sentence to further improve the fluency of the generated sentences. In contrast to the uni-modal dependency tree widely used in dependency parsing for language analysis, our multi-modal dependency tree effectively integrates both the visual and linguistic information, improving the relevance of the output sentence to the visual input.

To effectively optimize the captioning model, we propose a tree-structured reinforcement learning algorithm and a novel node-level reward tailor-made for the tree construction process. By evaluating the consistency between the parent-child node pairs in the constructed tree and those in the ground truth trees, the tree node-level reward enables the model to capture the topological structure of the ground-truth dependency trees. Compared to the sequence-level rewards in existing methods, our node-level reward recognizes the contribution of each node to the multi-modal dependency tree and avoids the reward ambiguity problem [8].

To evaluate the effectiveness of the proposed method, we conduct experiments on two difficult video captioning datasets, the ActivityNet Captions dataset and Charades Captions dataset. We also perform experiments on two most widely-used datasets, namely the MSVD dataset and MSR-VTT dataset. Compared to MSVD and MSR-VTT, the sentences in the ActivityNet Captions dataset and Charades Captions dataset are typically longer and describe more complex activities in the videos. The experiments on these datasets demonstrate that our method not only generates long and complex sentences with high fluency and relevancy, but is also effective for relatively simple sentences.

The main contributions of this paper are as follows:

- We propose a multi-modal dependency tree construction method for video captioning. With the help of tree topology, our model generates more fluent and relevant sentences by effectively capturing the syntactic and semantic dependencies in natural language, especially when generating long and complex sentences.
- We develop a novel tree-structured reinforcement learning algorithm that optimizes the captioning model using a node-level reward, which facilitates learning the topology of the dependency trees and alleviates the reward ambiguity problem.

## 2 Related Work

### 2.1 Video Captioning

Thanks to the recent advances in computer vision and natural language processing, many video captioning methods based on the encoder-decoder framework have been proposed. The pioneers of video captioning methods [9] employ a convolutional neural network (CNN) to extract visual features and a recurrent neural network (RNN) to generate the words in the sentences in a sequential manner. To further improve the performance of video captioning, some methods [10, 11] apply the attention mechanism in video captioning, which enables the model to focus on different temporal segments when generating the sentences. Inspired by the advances in natural language processing, some methods [12, 13] propose to learn better syntax representations with the help of part-of-speech

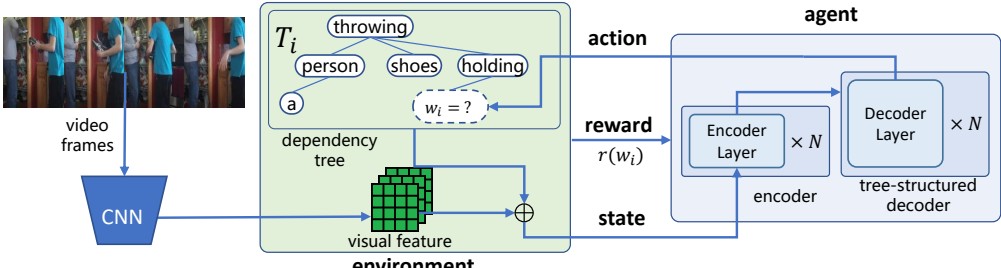

Figure 2: Overview of the proposed multi-modal dependency tree generation framework and the tree-structured reinforcement learning training algorithm. $T_i$ and $w_i$ denote the multi-modal dependency tree and the predicted word in the $i$-th step, respectively. The green-colored box in the middle stands for the *environment*, and the *state* of the environment is comprised of the perviously generated tree and the visual features. The blue-colored box on the right denotes the *agent*, namely the captioning model. During training, the agent executes an *action* by predicting a word $w_i$ in the $i$-th step, and receives a reward $r(w_i)$ from the environment.

tags. In order to provide richer semantic information to video captioning models, Rahman *et al.* [14] attempt to utilize the audio information.

Motivated by the advances in neural machine translation, several recent methods [3, 4, 6] investigate the application of Transformer [7] to video captioning, which uses multi-head self-attention mechanism to yield more expressive representations of the input. Some methods [15, 16] focus on optimizing video captioning models using reinforcement learning, where the rewards are calculated based on the n-gram statistics.

Instead of directly generating a word sequence, our method generates a sentence by first constructing a dependency tree and then traversing the tree. Compared to the sequence generation models, our method explicitly models the grammatical structure with tree topology and better preserves the semantic relationship by incorporating both visual and linguistic information into the dependency tree.

## 2.2 Tree-structured Language Generation

In recent years, tree-structured language generation methods are proposed for various natural language processing tasks, including program generation [17], generating math equation [18] and machine translation [19]. Alvarez *et al.* [17] generate the tree nodes in depth-first search order using doubly-recurrent neural network, which combines the hidden states from the parent node and the last sibling node when predicting the label for a new node. A dependency tree generation method is proposed by [19] to generate target sentences in machine translation. Instead of directly constructing the tree structure, this method generates canonizalized ternary trees where each node has a fixed number of child nodes. Liu *et al.* [18] propose to generate abstract syntax trees of math expressions to solve math problems described in natural language.

The work closest to our method is [20] that generates image captions by constructing canonicalized dependency trees. Different from this method that generates trees with only linguistic information, our method constructs multi-modal dependency trees that contain both visual and linguistic information to capture richer contextual information.

## 3 Our Method

### 3.1 Overview

In this section, we introduce the proposed multi-modal dependency tree generation framework. The training data is formulated as $D = \{(x_j, T_j)|_j\}$, where $x_j$ denotes the $j$-th video, and $T_j$ denotes the dependency tree of the corresponding sentence. Our model follows the encoder-decoder paradigm, where both the encoder and the decoder are stacks of $N$ self-attention layers. Given an input video, the encoder of our model takes a set of $d$-dimensional visual features $F = \{\boldsymbol{f}_1, \boldsymbol{f}_2, ..., \boldsymbol{f}_M\}$ as input,

where $\boldsymbol{f}_i \in \mathbb{R}^d$. The decoder firstly generates a multi-modal dependency tree $T = \{U, E\}$, where $U$ and $E$ denote the node set and edge set, respectively. Each node $u_i \in U$ corresponds to a word in the sentence, and a directed edge $\langle u_j, u_i \rangle \in E$ indicates that the node $u_i$ is dependent on its parent node $u_j$. Then we use a recursive algorithm to traverse the tree $T$ and recover the order of the words in the output sentence $y = \{w_1, w_2, ..., w_L\}$, where $L$ denotes the length of the sentence and $w_i$ denotes the $i$-th word. The overview of our framework is illustrated in Figure 2.

## 3.2 Tree Generation Process

Our tree generation model follows a step-by-step generation process. In the $i$-th step, a node $u_i$ is added to the tree, and the corresponding word $w_i$ is predicted. Formally, given a video $x$, the probability of generating a tree $T$ is given by

$$p(T|x) = \prod_{u_i \in T} p(u_i|x, T_{i-1}), \tag{1}$$

where $T_{i-1}$ denotes the sub-tree that has been generated in the $(i-1)$-th step.

Since the structure of the dependency tree is unknown in the tree generation process, our model also predicts the topological information related to each node. For each node $u_i$, we define three topological labels: $t_i^s \in \{0, 1\}$, $t_i^c \in \{0, 1\}$ and $t_i^e \in \{0, 1\}$. $t_i^s$ indicates whether node $u_i$ has another sibling node. If $t_i^s = 1$, another sibling node of $u_i$ will be added to the tree in the following steps. $t_i^c$ decides whether $u_i$ have child nodes, and least one child node of $u_i$ will be added when $t_i^c = 1$. $t_i^e = 0$ indicates that the node $u_i$ is the left child of its parent node $u_j$, namely the word $w_i$ appears on the left of $w_j$ in the output sentence, and $t_i^e = 1$ indicates that $u_i$ is the right child of $u_j$.

## 3.3 Encoder

Given a set of visual features $F = \{\boldsymbol{f}_1, \boldsymbol{f}_2, ..., \boldsymbol{f}_M\}$, the encoder uses multiple self-attention operations to capture the relationships between video frames represented by the visual features. We first recall the scaled dot-product attention used by the self-attention layers, which operates on three sets of vectors $\boldsymbol{Q}$, $\boldsymbol{K}$ and $\boldsymbol{V}$:

$$\text{Attention}(\boldsymbol{Q}, \boldsymbol{K}, \boldsymbol{V}) = \text{softmax}(\frac{\boldsymbol{Q}\boldsymbol{K}^\top}{\sqrt{d}})\boldsymbol{V}, \tag{2}$$

where $\boldsymbol{Q} \in \mathbb{R}^{n_q \times d}$ is a matrix consisting of $n_q$ query vectors, $\boldsymbol{K} \in \mathbb{R}^{n_k \times d}$ and $\boldsymbol{V} \in \mathbb{R}^{n_k \times d}$ are matrices containing $n_k$ key vectors and $n_k$ value vectors, respectively. The vectors in $\boldsymbol{Q}$, $\boldsymbol{K}$ and $\boldsymbol{V}$ are with the same dimensionality $d$. In addition to the self-attention layer, each encoder layer contains a feed-forward network (FFN) consisting of two fully connected layers, which can be formulated as

$$\text{FFN}(\boldsymbol{X}) = \text{ReLU}(\boldsymbol{X}\boldsymbol{W}_1 + \boldsymbol{b}_1)\boldsymbol{W}_2 + \boldsymbol{b}_2, \tag{3}$$

where $\boldsymbol{W_1}, \boldsymbol{W_2} \in \mathbb{R}^{d \times d}$ and $\boldsymbol{b}_1, \boldsymbol{b}_2 \in \mathbb{R}^d$ are learnable parameters.

In each layer of the encoder, the queries, keys and values are obtained by linearly mapping the input features, and the output of the $n$-th encoding layer $\boldsymbol{X}_n$ is given by

$$\boldsymbol{X}'_n = \text{Attention}(\boldsymbol{W}_n^q \boldsymbol{X}_{n-1}, \boldsymbol{W}_n^k \boldsymbol{X}_{n-1}, \boldsymbol{W}_n^v \boldsymbol{X}_{n-1}), \boldsymbol{X}_n = \text{FFN}(\boldsymbol{X}'_n), \tag{4}$$

where $\boldsymbol{W}_n^q \in \mathbb{R}^{d \times d}$, $\boldsymbol{W}_n^k \in \mathbb{R}^{d \times d}$ and $\boldsymbol{W}_n^v \in \mathbb{R}^{d \times d}$ are learnable parameters, and $\boldsymbol{X}_{n-1}$ is the output of the previous encoding layer. FFN represents the feed-forward network defined in Eq. 3. The visual features $V$ are used as the input of the first encoding layer.

## 3.4 Tree Structured Decoder

The decoder is conditioned on both the output of the encoder and the previously generated sub-tree. In the $i$-th step, the decoder takes the encoded visual feature $\boldsymbol{X}_N$, the words in the sub-tree $T_{i-1}$ and the multi-modal embeddings of the nodes in $T_{i-1}$ as input, and predicts the word $w_i$ together with the topological labels $t_i^s$, $t_i^c$ and $t_i^e$. The tree generation process terminates if the predicted word $w_i$ is a special token $\langle eos \rangle$ indicating the end of the sentence. Since the order of words may affect the semantics of the sentence, in each step, we traverse the tree $T_{i-1}$ using the algorithm that will be

described in Section 3.5 and obtain a node sequence $u_{k_1}, u_{k_2}, ..., u_{k_{i-1}}$ before feeding the words that have been generated so far to the decoder, where $k_i$ indicates the index of the nodes.

Let $\boldsymbol{E}_{i-1} \in \mathbb{R}^{(i-1) \times d}$ denote the embeddings of the previously generated $(i-1)$ words. The decoder acquires the multi-modal embedding of the current node $\boldsymbol{a}_i$ by firstly attending to the visual features and the embeddings of the previously generated words using self-attention layers, and then fusing the output of the self-attention. The $n$-th decoding layer is formulated as:

$$
\begin{aligned}
\boldsymbol{Y}_n &= \text{FFN}(\boldsymbol{Y}_n^{vis} + \boldsymbol{Y}_n^{word}) \\
&= \text{FFN}(\text{Attention}(\boldsymbol{Y}_{n-1}, \boldsymbol{X}_N, \boldsymbol{X}_N) + \text{Attention}(\boldsymbol{Y}_{n-1}, \boldsymbol{E}_{i-1}, \boldsymbol{E}_{i-1})).
\end{aligned}
\tag{5}
$$

$\boldsymbol{Y}_N \in \mathbb{R}^d$ represents the output of the last decoding layer and is used as the multi-modal embedding of the current node $u_i$, i.e. $\boldsymbol{a}_i = \boldsymbol{Y}_N$ in the $i$-th step. The probability distribution of the word corresponding to $u_i$ is calculated using the multi-modal embedding:

$$
\boldsymbol{p}_{w_i} = \boldsymbol{W}_{word} \boldsymbol{a}_i,
\tag{6}
$$

where $\boldsymbol{W}_{word} \in \mathbb{R}^{|D| \times d}$ is a learnable parameter, and $|D|$ denotes the vocabulary size. After the word $w_i$ is determined by a sampling strategy (e.g. greedy sampling or beam search), the topological labels of the current node are predicted by

$$
\begin{aligned}
p(t_i^s = 1) &= \text{sigmoid}(\boldsymbol{w}_s^\top [\boldsymbol{a}_i; \boldsymbol{E}_{w_i}]), \\
p(t_i^c = 1) &= \text{sigmoid}(\boldsymbol{w}_c^\top [\boldsymbol{a}_i; \boldsymbol{E}_{w_i}]), \\
p(t_i^e = 1) &= \text{sigmoid}(\boldsymbol{w}_e^\top [\boldsymbol{a}_i; \boldsymbol{E}_{w_i}]),
\end{aligned}
\tag{7}
$$

where $\boldsymbol{w}_s, \boldsymbol{w}_c, \boldsymbol{w}_e \in \mathbb{R}^{2d}$ are learnable parameters, $\boldsymbol{E}_{w_i} \in \mathbb{R}^d$ denotes the embedding of word $w_i$, respectively. The operator $[;]$ denotes the vector concatenation.

### 3.5 Tree Traversal Algorithm

We restore the word order and convert the tree to human-readable word sequence by traversing the tree. Given a generated dependency tree $T$, our tree traversal algorithm recursively processes each node in the tree, and restores the word order of the phrase represented by the node and its sub-trees. Since the tree is generated following the depth-first search, the generation order of the child nodes of each node is the same as the order of the corresponding words in the sentence. Thus, for an arbitary node $u_i$ in the dependency tree $T$, we sequentially traverse its left child nodes, the node $u_i$ and its right child nodes.

### 3.6 Model Training

#### 3.6.1 Pre-Training Stage

The training process of the multi-modal dependency generation model involves a pre-training stage and a fine-tuning stage. Since the model predicts both the word and the topology labels, we use a word loss $\mathcal{L}_w$ and a topological loss $\mathcal{L}_t$ in the pre-training stage. Formally, the word loss is defined as

$$
\mathcal{L}_w = -\sum_{i=1}^{|T|} \log p(w_i | T_{i-1}, x),
\tag{8}
$$

where $p(w_i | T_{i-1}, x)$ denotes the probability of predicting word $w_i$ given the previously generated tree $T_{i-1}$ and the visual input $x$, and $|T|$ denotes the number of nodes in the tree $T$. The topological loss is defined as

$$
\begin{aligned}
\mathcal{L}_t &= \mathcal{L}_{bce}(t_i^s, \hat{t}_i^s), + \mathcal{L}_{bce}(t_i^c, \hat{t}_i^c) + \mathcal{L}_{bce}(t_i^e, \hat{t}_i^e), \\
\mathcal{L}_{bce}(t, \hat{t}) &= -\sum_{i=1}^{|T|} \hat{t} \cdot \log(p(t)) + (1 - \hat{t}) \cdot \log(1 - p(t)),
\end{aligned}
\tag{9}
$$

where $\hat{t}_i^s$, $\hat{t}_i^c$ and $\hat{t}_i^e$ are ground-truth topological labels and $\mathcal{L}_{bce}$ denotes the binary cross-entropy loss. The overall loss function in the pre-training stage is formulated by

$$
\mathcal{L}_{pretrain} = \mathcal{L}_w + \mathcal{L}_t
\tag{10}
$$

### 3.6.2 Fine-tuning Stage

To effectively optimize the proposed model, we formulate the multi-modal dependency tree generation as a decision-making process and fine-tune the model with reinforcement learning. Specifically, the captioning model is regarded as the *agent*, the input video $x$ and the generated dependency tree $T_{i-1}$ are regarded as the *state* of the environment, and the prediction of the word $w_i$ is regarded as the *action*. Instead of using the same sequence-level reward for all the actions, we design a novel node-level reward that estimates the contribution of each individual action. The reward $r(w_i)$ for the word $w_i$ is given by

$$r(w_i) = \lambda_1(\text{CIDEr}(T_i) - \text{CIDEr}(T_{i-1})) + \lambda_2 \mathbb{1}(\langle u_i, u_p \rangle \in E), \tag{11}$$

where $\text{CIDEr}(T)$ denotes the CIDEr score of the word sequence represented by the nodes in $T$, $u_p$ denotes the parent node of $u_i$ and $\mathbb{1}(\langle u_i, u_p \rangle \in E)$ indicates whether the edge $\langle u_i, u_p \rangle$ is present in the ground-truth dependency tree. $\lambda_1$ and $\lambda_2$ are tunable hyper parameters. The discounted future reward for the agent is calculated by

$$R(w_i) = r(w_i) + \sum_{k=1}^{\infty} \gamma^k r(w_{i+k}), \tag{12}$$

where $\gamma$ denotes the discount factor. Let $\theta$ denote the parameters of the captioning model, and then the loss function in the fine-tuning stage is given by

$$\mathcal{L}_{rl} = -\mathbb{E}_{w_i \sim \pi(\theta)}(R(w_i)), \tag{13}$$

where $\pi(\theta)$ denotes the policy defined by $\theta$.

## 4 Experiments

### 4.1 Datasets

**ActivityNet Captions** is a dense video captioning dataset that contains 10,030 training videos, 4,926 validation videos and 5,044 test videos. Each video is annotated with an average of 3.65 sentences together with temporal annotation and the average length of the sentences is 13.7 words. We conduct the experiments on ActivityNet Captions with ground-truth proposals and the results are reported on the validation set.

**Charades Captions** is composed of 9,223 videos of indoor activities. Each video is annotated with 2-5 sentences, and the average length of the sentences is 24.13 words. Following [15], we split this dataset into 6,963 training videos, 500 validation videos and 1,760 test videos.

**MSVD** consists of 1,970 video clips collected from YouTube, each of which annotated with about 41 sentences and the average length of the sentences is 7.10 words. We follow [9] to split the dataset into 1,200 training videos, 100 validation videos and 670 testing videos.

**MSR-VTT** is a dataset collected for open-domain video captioning, which contains 10,000 video clips in total. Each video in the MSR-VTT dataset contains 20 human annotated captions. We use the splits provided by [15], where the training split, validatation split and test split contain 6,513 videos, 497 videos and 2,990 videos, respectively. The average length of the sentences in MSR-VTT is 9.28 words.

### 4.2 Evaluation Metrics

We report several widely-used automatic evaluation metrics for video captioning, including Bleu-n [21], METEOR [22], ROUGE-L [23] and CIDEr [24]. Nevertheless, these n-gram based evaluation metrics have some limitations, e.g., they penalize the phrases that are semantically correct but differ from ground-truth in the specific word choices [25]. To evaluate the quality of the sentences more effectively, we also report Improved BERTScore metric proposed by [26]. Compared with the n-gram based metrics, Improved BERTScore is computed using pre-trained BERT embeddings and has better correlation with human experts.

## 4.3 Implementation Details

In the training process, in order to convert the sentences to dependency trees, we use the dependency parser in the SpaCy toolkit [27]. For the ActivityNet Captions dataset, we extract the visual features of the 16-frame video segments using the pre-trained C3D network. To compare our method with [3], we also extract the frame features using ResNet-200 [28], and extract the optical flow features using BN-Inception [29]. For the Charades Captions dataset, we sample the video frames at 3 fps and extract the feature of each frame using the pre-trained ResNet-152 network [28] following [15]. We use the Adam optimizer [30] in both pre-training and fine-tuning stages. The initial learning rate is set to $5 \times 10^{-5}$ and decays 0.8 times for every 3 epochs. The experiments are conducted using one NVIDIA RTX 2080Ti GPU.

Table 1: Comparison with state-of-the-art methods and the results of ablation studies on the ActivityNet Captions dataset using ground-truth proposals. B@n, M, C and BS are abbreviations for Bleu-n, METEOR, CIDEr and Improved BERTScore, respectively. *Note that the last two rows are compared for different video features.*

| Feature | Model | B@1 | B@2 | B@3 | B@4 | M | C | BS |
|---|---|---|---|---|---|---|---|---|
| | DCE [31] | 18.13 | 8.43 | 4.09 | 1.60 | 8.88 | 25.12 | - |
| | DVC [32] | 19.57 | 9.90 | 4.55 | 1.62 | 10.33 | 25.24 | - |
| | SDVC [33] | 28.02 | 12.05 | 4.41 | 1.28 | 13.07 | 43.48 | - |
| | WLT [14] | 15.23 | 6.58 | 3.04 | 1.46 | 7.23 | 25.36 | - |
| C3D | w/o tree | 27.01 | 11.12 | 4.10 | 1.47 | 12.40 | 42.30 | 35.57 |
| | w/o RL | 20.31 | 8.54 | 3.90 | 1.40 | 11.23 | 38.90 | 34.30 |
| | w/o tree, RL | 20.56 | 9.45 | 4.48 | 1.26 | 9.97 | 37.80 | 30.20 |
| | w/o visual embedding | 25.30 | 10.90 | 3.74 | 1.36 | 11.09 | 35.10 | 32.69 |
| | sequence reward | 27.35 | 11.18 | 4.15 | 1.62 | 11.70 | 41.39 | 35.23 |
| | Ours | **28.53** | **12.12** | **4.49** | **1.75** | **13.24** | **44.13** | **36.35** |
| ResNet-200+ | Masked [3] | 23.93 | 12.16 | **5.76** | 2.71 | 11.16 | 47.71 | - |
| BNInception | Ours | **24.10** | **12.26** | 5.57 | **2.72** | **11.20** | **47.80** | - |

Table 2: Experiment results and the results of ablation studies on the Charades Captions dataset. R is the abbreviation for ROUGE-L.

| Model | B@1 | B@2 | B@3 | B@4 | M | R | C | BS |
|---|---|---|---|---|---|---|---|---|
| HRL [15] | 64.40 | 44.30 | 29.40 | 18.80 | 19.50 | 41.40 | 23.20 | - |
| w/o tree | 62.10 | 43.50 | 28.49 | 17.68 | 18.65 | 40.20 | 23.50 | 34.13 |
| w/o RL | 58.40 | 39.30 | 25.31 | 17.08 | 18.19 | 38.97 | 22.53 | 33.71 |
| w/o tree, RL | 57.63 | 38.32 | 24.83 | 16.50 | 18.21 | 38.80 | 21.45 | 30.41 |
| w/o visual embedding | 60.39 | 40.30 | 26.70 | 16.90 | 18.19 | 39.45 | 21.87 | 32.50 |
| sequence reward | 64.00 | 44.90 | 29.70 | 18.50 | 19.20 | 40.32 | 23.41 | 34.21 |
| Ours | **65.30** | **45.60** | **29.80** | **18.87** | **19.38** | **40.50** | **24.21** | **34.57** |

## 4.4 Comparison with State-of-the-Art Methods

The results on ActivityNet Captions are shown in Table 1. We compare our method with DCE [31], DVC [32], SDVC [33], and Masked (which uses different video features) [3]. All these methods generate sentences using sequence models. The captioning models in [32, 33, 14] are implemented by RNN and the model proposed by [3] is based on Transformer.

All the results of these compared methods are directly quoted from their original papers. Our method achieves the best performance on the ActivityNet Captions dataset, demonstrating that by explicitly modeling the hierarchical structure of the sentences, our proposed method can generate long and complex sentences with higher fluency and relevance.

We compare our method with HRL [15] on the Charades Captions dataset. This method involves a manager network that designs sub-goals and a worker network that fulfills each sub-goal by predicting words, both of which are trained end-to-end using hierarchical reinforcement learning. The results are shown in Table 2. We observe that our method outperforms HRL, which evaluates the effectiveness of our method in generating complex video descriptions.

The results on the MSVD dataset and the MSR-VTT dataset are shown in Table 3. We compare our method with HRL [15], MARN [34], POS-CG [13], Joint [12] and RMN [35].All these methods generate sentences using sequence models implemented by RNN. All the results of these compared methods are directly quoted from their original papers. For fair comparison, we use the same visual features as these methods. From the results on MSVD and MSR-VTT, we observe that our method achieves comparable performance with the state-of-the-art methods when using different visual features, which demonstrates that our method is also effective for generating relatively simple sentences. Note that in such simple cases our proposed method is not expected to make much difference or perform better.

Table 3: Comparison with state-of-the-art methods on MSVD and MSR-VTT using different visual features. I3D(RGB) and I3D(OF) indicate using I3D network [36] to extract the features of raw video frames and optical flow, respectively. The mark * denotes using additional object features.

| Feature | Model | MSVD | | | | MSR-VTT | | | |
|---|---|---|---|---|---|---|---|---|---|
| | | B@4 | R | M | C | B@4 | R | M | C |
| ResNet152 | HRL [15] | - | - | - | - | 41.3 | 61.7 | 28.7 | 48.0 |
| | Joint [12] | 52.1 | 69.8 | 33.7 | 80.6 | 41.4 | **62.0** | 28.9 | 48.1 |
| | Ours | **52.2** | **70.0** | **34.0** | **81.2** | **41.6** | 62.0 | **29.1** | **48.4** |
| ResNet101+ | MARN [34] | 48.4 | 71.9 | 35.1 | 92.2 | **40.4** | 60.7 | 28.1 | 47.1 |
| ResNext101 | Ours | **49.0** | **72.2** | **35.3** | **92.5** | 40.2 | **61.1** | **28.2** | **47.3** |
| InceptionResnetV2+ | POS-CG[13] | **52.5** | 71.3 | 34.1 | 92.0 | **42.0** | **61.6** | 28.2 | 48.7 |
| I3D(OF) | Ours | 51.7 | **71.6** | **34.9** | **92.4** | 41.8 | 61.4 | **28.3** | **49.0** |
| InceptionResNet+ | RMN (H+L)*[35] | **54.6** | **73.4** | 36.5 | **94.4** | 42.5 | 61.6 | 28.4 | 49.6 |
| I3D(RGB) | RMN [35] | 52.5 | 72.7 | 36.1 | 92.8 | 42.5 | 61.6 | 28.4 | 49.6 |
| | Ours | 51.8 | 72.8 | **36.7** | 92.5 | **42.5** | **61.8** | **29.1** | **49.8** |
| InceptionResNet+ | ORG-TRL*[37] | **54.3** | **73.9** | 36.4 | **95.2** | **43.6** | **62.1** | 28.8 | **50.9** |
| C3D | PMI-CAP[38] | 54.7 | - | 36.4 | 95.2 | 42.2 | - | 28.8 | 49.5 |
| | Ours | 52.0 | 72.8 | **36.5** | 92.6 | 42.7 | 61.9 | **29.4** | 50.1 |

## 4.5 Ablation Studies

To analyze the effect of different components, we conduct ablation studies on the ActivityNet Captions dataset and the Charades Captions dataset. The following variants of our method are evaluated:

- **w/o tree**: To evaluate the advantage of the tree structure over the sequence structure, we replace the dependency trees with chain-structured trees, where the root node corresponds to the leftmost word in the sentence. For each non-leaf node, its only child node corresponds to the word on its right. This variant of our model is actually a sequence model that generates the words in the sentence in left-to-right order.

- **w/o RL**: To validate the effectiveness of reinforcement learning, we only optimize the model with the cross-entropy loss in pre-training stage.

- **w/o tree, RL**: To evaluate the effect of using tree structure and reinforcement learning simultaneously, we optimize a sequence model with only cross-entropy loss.

- **w/o visual embedding**: To evaluate the effectiveness of the visual embeddings of the tree nodes, we replace the attended visual feature $Y_n^{vis}$ defined in Eq. 5 with an all-zero matrix, and the node representations contains only linguistic information.

- **sequence reward**: To verify the contribution of the tree node-level reward, we replace it with the sequence-level reward, i.e. the reward of each node equals to the CIDEr score of the entire sentence.

The results of ablation studies on the ActivityNet Captions and Charades Captions datasets are shown in the middle part of Table 1 and Table 2, respectively. From these results, we make the following observations. First, by replacing the dependency tree with the chain-structured tree, the performance of our model degrades in terms of all the metrics, indicating that the hierarchical structure of the sentences is beneficial for generating sentences with high quality. Second, our full model outperforms "w/o visual embedding", validating the superiority of the multi-modal representation of the tree node by taking full advantage of both visual and linguistic information. Third, when the fine-tuning stage

with reinforcement learning is removed, our model performs worse on all the metrics, which validates the contribution of our reinforcement learning algorithm. Our full model also outperforms "sequence reward", which verifies that the node-level reward effectively guides the training process.

## 4.6 Evaluation on Simple and Complex Subsets

To further evaluate the effect of the tree structure on generating simple sentences and complex sentences, we split the videos in the test sets of MSR-VTT and Charades Captions into a simple subset and a complex subset according to the average length of the ground-truth sentences. The data distributions of the two subsets and the evaluation results are shown in Table 4. From these results, we observe that by utilizing the tree structure, the performance of our model significantly increases on the complex subset, which demonstrates the proposed model's capability in terms of generating complex sentences.

Table 4: Data distribution and evaluation results on the simple subset and complex subset of Charades Captions and MSR-VTT.

| Subset | Sentence length | Video number | w/o tree | | | Ours | | |
|---|---|---|---|---|---|---|---|---|
| | | | B@4 | R | C | B@4 | R | C |
| Charades (simple) | ≤ 20 | 550 | 18.8 | 36.7 | 28.4 | 17.9 | 40.9 | 25.8 |
| Charades (complex) | > 20 | 1210 | 17.1 | 33.3 | 20.8 | 18.8 | 39.8 | 24.4 |
| MSR-VTT (simple) | ≤ 10 | 2005 | 38.3 | 62.3 | 54.8 | 45.8 | 66.2 | 59.8 |
| MSR-VTT (complex) | > 10 | 985 | 31.0 | 48.6 | 25.3 | 35.1 | 50.8 | 27.7 |

## 4.7 Results of Using Different Evaluation Metric in the Reward

To assess the impact of the evaluation metric used in the reward on the performance of the captioning model, we conduct additional experiments that replace the CIDEr score in the reward using other metrics (i.e. Bleu-4 and ROUGE-L) on the Charades Captions dataset. From the results in Table 5, we observe that when using Bleu-4 (or ROUGE-L) in the reward, the performance in terms of Bleu-4 (or ROUGE-L) increases when compared with using CIDEr in the reward. However, using CIDEr score in the reward leads to the best overall performance. Thus, we use CIDEr score in the reward in other experiments.

Table 5: The results of using different evaluation metrics in the reward on the Charades Captions dataset.

| Method | Reward | B@4 | M | R | C |
|---|---|---|---|---|---|
| Ours | Bleu-4 | 19.02 | 18.50 | 40.53 | 22.95 |
| Ours | ROUGE-L | 17.06 | 18.32 | 41.79 | 22.32 |
| Ours | CIDEr | 18.87 | 19.38 | 40.50 | 24.21 |

## 4.8 Qualitative Results

We show some examples of the generated sentences on Charades Captions and MSR-VTT in Figure 3. As shown in the figure, by utilizing the tree structure, our method generates sentences that not only describe the objects and the human actions more accurately but also possess correct grammatical structures.

## 4.9 Human Evaluation

To intuitively evaluate the effect of the tree structure on the quality of the generated sentences, we conduct human evaluation on the test splits of ActivityNet Captions and MSR-VTT. We randomly choose 200 videos from the test splits of ActivityNet Captions and MSR-VTT, respectively. The workers are given the original video as well as two sentences generated by "Ours" and "w/o tree", and are asked to select the sentence that has better relevance to the video and the sentence with better fluency. For the two methods "Ours" and "w/o tree", we report the percentage of sentences that have better relevance and the percentage of sentences that have better fluency. From the results in Table

| | **Ours:** A person is standing in front of a mirror holding a towel. The person puts the towel on a shelf and leaves. | **w/o tree:** A person is holding a book and a phone. The person puts the book on the floor and begins tidying up the room. | **gt:** A person walks to a mirror and begins wiping it with a towel. The person then tosses the towel on a couch and adjusts a table. |
|---|---|---|---|
| | **Ours:** A person is sitting on the floor reading a book. The person puts the book on a shelf picks up a book and leaves. | **w/o tree:** A person is sitting on the floor reading a book while reading a book. | **gt:** She is sitting laundry room and reading book, and now she is lying on floor while reading, and put the sheet under her head. |
| | **Ours:** A man is talking about space. | **w/o tree:** There is a man talking about something. | **gt:** A man talks about the benefits of defensive satellites. |
| | **Ours:** A person is playing a golf game. | **w/o tree:** A man is in a green shirt playing a baseball game. | **gt:** A golf player is trying to hit the ball into the pit. |

Figure 3: Qualitative results on Charades Captions (the first two rows) and MSR-VTT (the last two rows). "gt", "Ours" and "w/o tree" denote the ground-truth, the sentence generated by our method and the sentence generated by the variant "w/o tree" of our model, respectively.

6, we observe that the tree structure remarkably improves both the relevance and the fluency of the generated sentences.

Table 6: The results of human evaluation.

| Dataset | Relevance | | Fluency | |
|---|---|---|---|---|
| | w/o tree | Ours | w/o tree | Ours |
| ActivityNet | 44.63% | 55.36% | 48.02% | 51.97% |
| MSR-VTT | 45.65% | 54.34% | 46.20% | 53.80% |

## 5   Conclusions

We have presented a multi-modal dependency tree generation method for video captioning. Our model can explicitly model the hierarchical structure of natural language using tree topology to better capture both the syntactic structure and semantic relationship in sentences. Both visual and linguistic information are incorporated into the node embeddings to further improve the relevance of the generated sentence to the video. Moreover, we have developed a tree-structured reinforcement learning algorithm and designed a novel tree node-level reward to effectively optimize the captioning model. Our multi-modal dependency tree is capable of generating complex sentences with high fluency and relevance for videos. Extensive experiment on multiple challenging video captioning datasets have demonstrated the effectiveness of our method.

## Funding Transparency Statement

This work was supported in part by the Natural Science Foundation of China (NSFC) under Grant No 62072041.

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
