# Supplementary Material for
# Multi-modal Dependency Tree for Video Captioning

**Wentian Zhao, Xinxiao Wu**[*]
Beijing Laboratory of Intelligent Information Technology
School of Computer Science
Beijing Institute of Technology
No. 5, Zhongguancun South Street, Beijing, China
{wentian_zhao,wuxinxiao}@bit.edu.cn

**Jiebo Luo**
Department of Computer Science
University of Rochester
Rochester, NY 14627 USA
jluo@cs.rochester.edu

## A   Broader Impact

This paper introduces a novel video captioning method that generates sentences by constructing dependency trees. The proposed method offers a possible new way of generating fluent and relevant sentences for videos and may inspire more works that explicitly model the syntactic structure of sentences in natural language generation. It can also help develop more practical video processing systems, such as automatic video subtitling tools. However, such technique is still affected by the biases in the training data. When the videos involve minorities or uncommonly-seen subjects, it may produce undesired output or lead to inaccurate understanding of the video content. Thus, more future research is necessary to address this issue.

## B   Qualitative Analysis of Generated Tree Structure

We show some examples of the generated sentences on the MSVD dataset and the Charades Captions dataset in Figure 1 and Figure 2, respectively. As is shown in these figures, our method describes the human actions accurately and the sentences possess correct grammatical structure. These qualitative results show that our tree topology improves the fluency and relevancy of sentences by effectively capturing the syntactic structure and semantic relationship.

## C   Automatic Evaluation of Generated Tree Structure

To evaluate the quality of the generated tree structure, we calculate the average edit distance between the generated tree and the ground truth dependency trees using the algorithm proposed in [1]. A lower edit distance indicates that the generated tree is more similar to the ground-truth dependency trees. The evaluation results on the Charades Captions dataset are shown in Table 2.

## D   Details about Simple and Complex Subsets

For the Charades Captions and MSR-VTT datasets, the videos in the complex subset and simple subset are selected from the test split according to the average ground-truth sentence length. A video is in the simple subset if the average length of its ground-truth sentences is less of equal than $p$, and is in the complex subset otherwise. The distributions of the average ground-truth sentence length on the test splits of Charades Captions and MSR-VTT are shown in Figure 3a and Figure 3b, respectively. According to the sentence length distributions, the value of $p$ on the Charades Captions dataset is set

---

[*]Corresponding author.

35th Conference on Neural Information Processing Systems (NeurIPS 2021), Sydney, Australia.

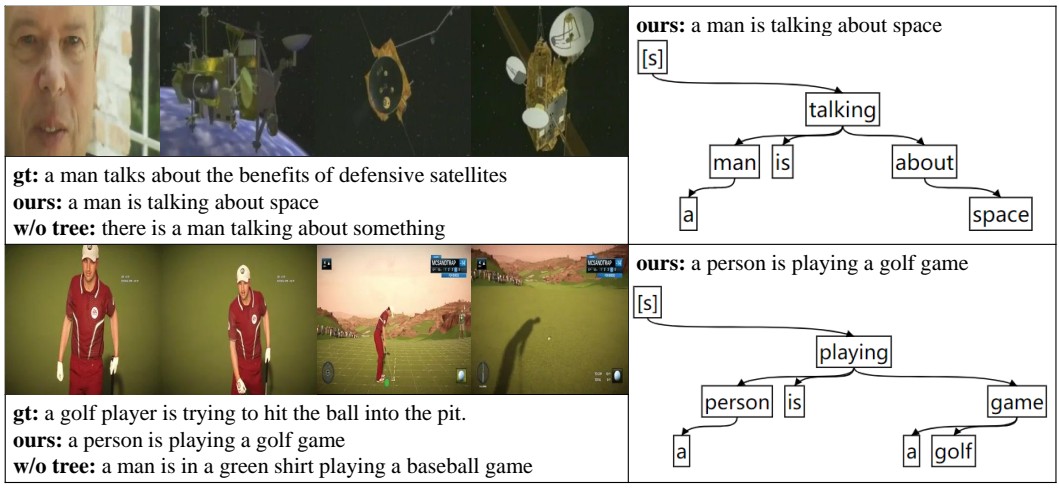

Figure 1: Qualitative results of the generated tree structure and sentences on the MSR-VTT dataset. "gt" and "ours" denote the ground-truth sentences and the output of our method, respectively. The sentences labeled with "w/o tree" are generated by the variant of our model that constructs chain-structured trees rather than dependency trees. The node with symbol $[s]$ on the top-left of the tree indicates the beginning of the sentence.

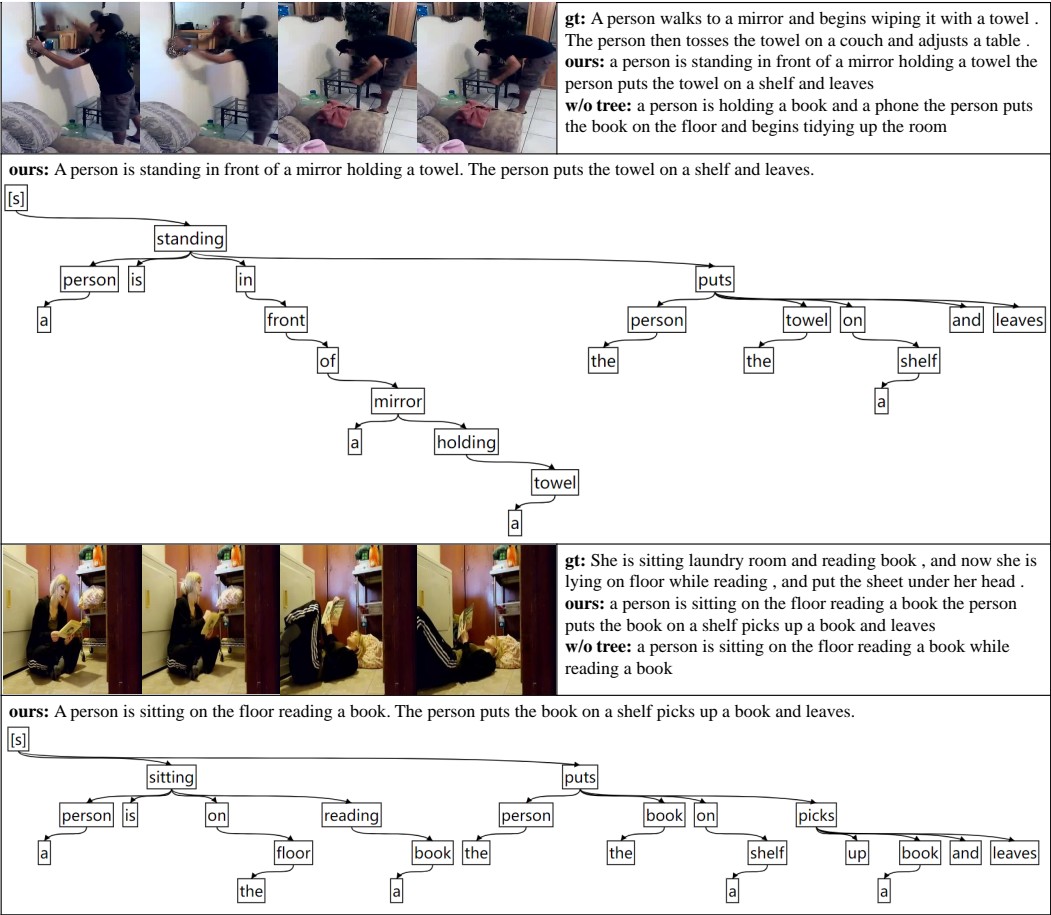

Figure 2: Qualitative results of the generated tree structure and sentences on the Charades Captions dataset.

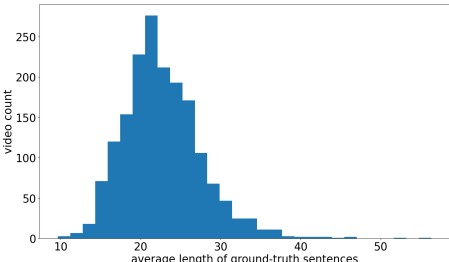 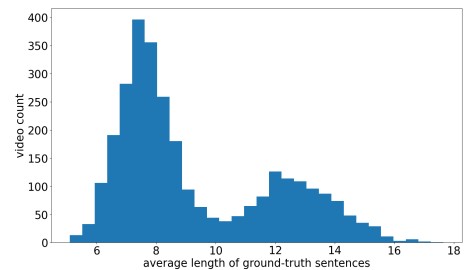

(a) Sentence length distribution on the test set of Charades Captions.

(b) Sentence length distribution on the test set of MSR-VTT.

Figure 3: Distribution of average ground-truth sentence lengths on the test sets of Charades Captions (a) and MSR-VTT (b).

to 20 and the value of $p$ on the MSR-VTT dataset is set to 10. As a result, the simple and complex subsets of Charades Captions contain 550 videos and 1,210 videos, respectively. The simple and complex subsets of MSR-VTT contain 2,005 videos and 985 videos, respectively.

We also calculated the average lengths of the ground-truth captions and the generated captions on the simple and complex subsets of Charades Captions and MSR-VTT, and the results are shown in Table 1.

Table 1: The average sentence lengths of ground-truth captions and the captions generated by "w/o tree" and "Ours" on the simple subset and complex subset of Charades Captions and MSR-VTT dataset.

| Subset | sentence lengths | | |
|---|---|---|---|
| | Ground-truth | w/o tree | Ours |
| Charades (all) | 22.710 | 18.443 | 21.729 |
| Charades (simple) | 17.780 | 17.591 | 21.370 |
| Charades (complex) | 25.071 | 18.853 | 21.894 |
| MSR-VTT (all) | 9.315 | 7.942 | 7.266 |
| MSR-VTT (simple) | 7.630 | 7.586 | 6.959 |
| MSR-VTT (complex) | 12.746 | 8.668 | 7.890 |

Table 2: The average tree edit distance of different variants of our method on the Charades Captions dataset. Lower average edit distance is better.

| model | average edit distance ($\downarrow$) |
|---|---|
| w/o visual embedding | 14.93 |
| w/o reinforcement learning | 15.05 |
| sequence reward | 14.76 |
| ours | 14.53 |

# E   More Results of Ablation Studies

To measure the significance of the results, we performed multiple 5 runs of the ablation studies on the Charades Captions dataset and report the mean $\pm$ standard deviation results and the results are shown in Table 3.

# F   Results of Image Captioning

To further validate the generalizability and scalability of the proposed method, we also conduct preliminary experiments of image captioning on MSCOCO [2] dataset. We use the output before

Table 3: The mean ± standard deviation of the ablation studies on Charades Captions dataset.

| Model | B@4 | M | R | C |
|---|---|---|---|---|
| w/o tree | 17.68±0.20 | 18.65±0.14 | 40.20±0.09 | 23.50±0.13 |
| w/o RL | 17.08±0.26 | 18.19±0.10 | 38.97±0.12 | 22.53±0.13 |
| w/o tree, RL | 16.50±0.19 | 18.21±0.12 | 38.80±0.12 | 21.45±0.12 |
| w/o visual embedding | 16.90±0.20 | 18.19±0.12 | 39.45±0.14 | 21.87±0.10 |
| sequence reward | 18.50±0.24 | 19.20±0.13 | 40.32±0.10 | 23.41±0.14 |
| Ours | 18.87±0.25 | 19.38±0.20 | 40.50±0.13 | 24.21±0.12 |

the last average pooling layer of ResNet-101 as the features of the images. From the results in Table 4, we observe that the proposed tree-structured decoding method can also be applied to the task of image captioning.

Table 4: Image captioning results on MSCOCO dataset.

| Method | B@4 | M | R | C |
|---|---|---|---|---|
| w/o tree | 36.26 | 27.21 | 56.38 | 119.52 |
| Ours | 37.03 | 27.87 | 56.42 | 120.30 |

## G  Details About Human Evaluation

In this section, we illustrate more details of human evaluation. We recruited 10 annotators to carry out the human evaluation process. Each annotator is given the original video, a ground-truth sentence and two sentences generated by "Ours" and "w/o tree" and is asked to compare the relevancy and fluency of the generated sentences. The user interface for human evaluation is shown in Figure 4. To ensure the fairness of human evaluation, among the two sentences generated by "Ours" and "w/o tree", we randomly choose one sentence to be "sentence 1" in the user interface and the other sentence is "sentence 2".

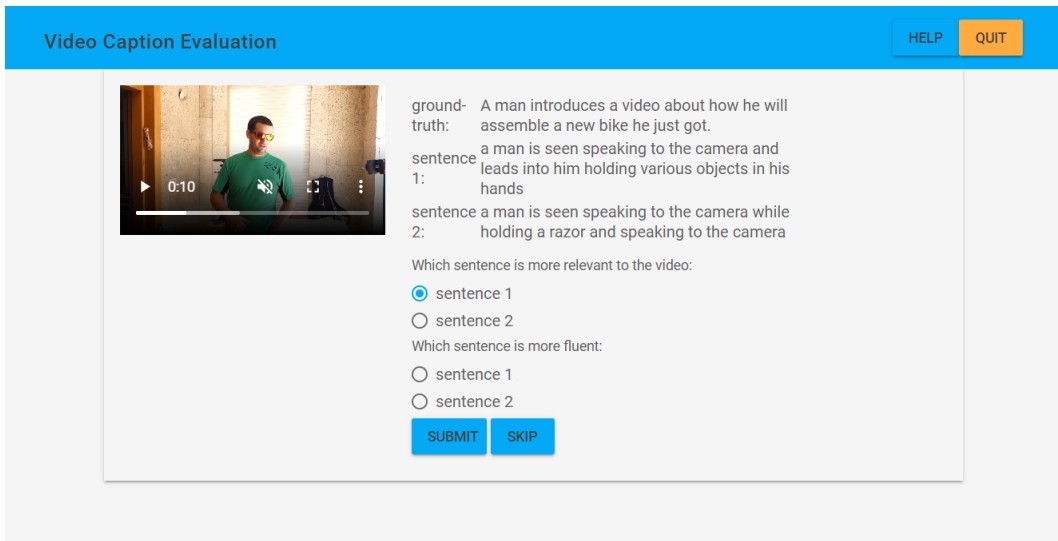

Figure 4: The graphical user interface for human evaluation.