# OpenReview forum: "Multi-modal Dependency Tree for Video Captioning"
_NeurIPS.cc/2021/Conference — NeurIPS 2021 Poster_

### Official Review · Reviewer_2Mir · 2021-07-03

**Rating:** 7
**Confidence:** 4

**Summary:**

The paper proposes a new architecture that combines a multimodal dependency tree approach with a tree-structured reinforcement learning method to improve upon existing methods on the video captioning task. The authors conduct experiments to understand the impact of the two components on four datasets. Experiments are made with varying visual features, on settings of varying complexity, and based on human evaluation. The results and analysis show improvements over the state of the art and that adopting a multimodal tree structure is a promising area of research for this task.

**Limitations And Societal Impact:**

I don't think there is a lot to add on negative societal impact since this paper is based on existing datasets and incremental contributions to the video captioning model. The error analysis I suggested before could help address some of the limitations and potentially surface societal problems related to the situations where the model fail but it is hard to make that claim before the initial error analysis is conducted.

**Main Review:**

Clairty:
The paper is clear and easy to follow despite multiple ideas being introduced. Images and the structured formulas are useful in understanding the concept and methodology is a significant part of the paper. One part that was a bit confusing on my first read was page 4 section 3.3, when the attention part is being described - Unless I misunderstood, the authors are using V as both the Visual features and the Value vectors. Although they might potentially be the same, it is unclear to me that this is always true since the authors also claim to use a multimodal representation per node. It might be better to use different variable names if they are not always the same or to clearly state that they will always be the same.

Originality:

By itself, the two contributions wouldn't much original. The multimodal tree structure would be too incremental if compared to previous work, mainly a mix of existing contributions. Also, incorporating CIDEr in the reward function has been tried before, although this specific reward function is different and incorporates the tree structure as well. That said, both contributions combined might make it sufficiently original for a publication.

Quality:
At first I was not convinced that introducing CIDEr in the reward function would be fair with the other methods that are not directly optimized for the metric but the improvements are constant throughout several different (although correlated) metrics. One improvement on this aspect might be to share how the results would change if CIDEr was replaced by one of the other metrics like Rouge and see the effect of that choice on the reward function - that might show how robust the reward is to the quality of the metric.

I would have liked to see significance scores for the results as well, I think these are lacking and might be relevant on some of the results that are too similar to each other. Also, for the ablation studies, it would be interesting to see the mean and standard deviation of multiple runs before making the claims the paper makes.

During the experiments for section 4.6, the w/o tree method performs better on one of the short and simple scenarios, an interesting finding that is not discussed by the authors and could potentially 1- expose limitations of the method and 2- inform future research on how to improve the proposed method.

For the Qualitative results like the ones proposed on 4.7, it would be interesting to see some error analysis and understand where the attention + tree based methods fails. For example, on the second result shown the captioning system says the person left the room while the person never leaves. Maybe an analysis of the attention components could show where the problem was - was it the attention, tree structure?

Overall, this is a well-written and interesting piece of work. I would appreciate more error analysis and significance tests to make it even stronger but I believe it is a good contribution to the video captioning task and reinforces directions related to reinforcement learning and tree-based text generation under this context.

**Time Spent Reviewing:**

2

---

> ### Author Response · Authors · 2021-08-10
> **Response to Reviewer 2Mir**
>
> We appreciate the strong support of Reviewer 2Mir and we answer the questions below.
>
> **Q1:** *One part that was a bit confusing on my first read was page 4 section 3.3, when the attention part is being described - Unless I misunderstood, the authors are using $V$ as both the Visual features and the Value vectors.*
>
> **A1:** Thank you for your helpful comments. The visual features that are used as the input of the encoder and the value vectors of the dot-product attention mechanism are different vectors. We have changed the notation of the visual features in the revised paper.
>
> **Q2:** *At first I was not convinced that introducing CIDEr in the reward function would be fair with the other methods that are not directly optimized for the metric but the improvements are constant throughout several different (although correlated) metrics. One improvement on this aspect might be to share how the results would change if CIDEr was replaced by one of the other metrics like Rouge and see the effect of that choice on the reward function - that might show how robust the reward is to the quality of the metric.*
>
> **A2:** We replace the CIDEr score in the reward using Bleu-4 and ROUGE-L, and the results on Charades Captions dataset are shown in the table below. We observe that compared with CIDEr reward, using ROUGE-L as rewards leads to inferior performance on all other metrics except ROUGE-L. Though using Bleu-4 as reward results in the highest Bleu-4 score, the performance on other metrics are lower than the model trained using CIDEr reward. It has also been validated in [Rennie et al. 2017] that using the CIDEr score as the reward increases the overall captioning performance most significantly.
>
> Method                  | B@4   | M     | R     | C
> --:|--:|--:|--:|--:|
> Ours (Bleu-4 reward)    | 19.03 | 18.50 | 40.53 | 22.97
> Ours (ROUGE-L reward)   | 17.06 | 18.32 | 41.79 | 22.32
> Ours                    | 18.90 | 19.80 | 41.70 | 24.20
>
> [Rennie et al. 2017] "Self-critical sequence training for image captioning." In CVPR 2017.
>
> **Q3:** *I would have liked to see significance scores for the results as well, I think these are lacking and might be relevant on some of the results that are too similar to each other. Also, for the ablation studies, it would be interesting to see the mean and standard deviation of multiple runs before making the claims the paper makes.*
>
> **A3:** Thank you for your constructive suggestion. During the rebuttal period, we run the ablation experiments for only 4 times on Charades Captions dataset due to limited time, and the mean $\pm$ standard deviation of are shown in the table below.
>
> | Model                | B@4               | M          | R          | C          |
> | -------------------- | ----------------- | ---------- | ---------- | ---------- |
> | w/o tree             | 17.68±0.20        | 18.65±0.14 | 40.20±0.09 | 23.51±0.13 |
> | w/o visual embedding | 16.90±0.30        | 18.19±0.12 | 39.45±0.14 | 21.87±0.10 |
> | w/o RL               | 16.18±0.26        | 18.19±0.10 | 38.97±0.12 | 21.54±0.13 |
> | sequence reward      | 18.52±0.24        | 19.20±0.13 | 41.60±0.10 | 23.52±0.14 |
> | Ours                 | 18.90±0.25        | 19.81±0.40 | 41.71±0.08 | 24.20±0.19 |
>
> **Q4:** *During the experiments for section 4.6, the w/o tree method performs better on one of the short and simple scenarios, an interesting finding that is not discussed by the authors and could potentially 1- expose limitations of the method and 2- inform future research on how to improve the proposed method.*
>
> **A4:** Thank you for your valuable suggestion. Our method tends to generate longer sentences even for simple videos, so it performs worse than "w/o tree" method in simple scenarios where the ground-truth sentences are short. In our future work, we are going to design a sentence length controlling mechanism that automatically adjusts the sentence length according to the complexity of the video content. We will add more analysis and discussions in the revised paper.
>
> **Q5:** *For the Qualitative results like the ones proposed on 4.7, it would be interesting to see some error analysis and understand where the attention + tree based methods fails. For example, on the second result shown the captioning system says the person left the room while the person never leaves. Maybe an analysis of the attention components could show where the problem was - was it the attention, tree structure?*
>
> **A5:** Thank you for your valuable suggestion. Such failure case may be caused by the prediction error of the topological labels that further affects the structure of the generated dependency tree. Taking the second qualitative result on Charades for example (the dependency tree is shown in Figure2 in the supplementary material), the topological label of the third-from-last node ("book") is predicted as 1, which leads to generating redundant sibling nodes ("and leaves").
> We will add more detailed analysis and discussions about the failure cases in the revised paper.

---

> > ### Comment · Reviewer_2Mir · 2021-09-10
> > **Thanks for the response. I am more confident about the previous score I gave.**
> >
> > Thank you for the detailed response. I was between a 6 and a 7 before as I was not convinced about some of the claims but I believe your answers guaranteed the 7 from my side. I appreciate the additional experiments and more complete analysis.

---

### Official Review · Reviewer_NzTM · 2021-07-15

**Rating:** 7
**Confidence:** 3

**Summary:**

This paper studies the task of video captioning. It proposes a structured approach where models generate a dependency tree for the caption, instead of just a sentence. The nodes of the dependency tree are words, and the tree is generated in depth-first order; thus, the sentence can be easily recovered by iterating through the tree.

The model is learned by a "pretraining stage"; where a left-to-right style loss is used (generating the next word, and the node in the tree, given previous nodes). In a "finetuning stage", the model is optimized using reinforcement learning with the improvement in CIDEr score as the reward (along with a reward as to whether the generated edge is in the ground truth tree or not).

**Ethical Concerns:**

None that come to mind to this reviewer.

**Limitations And Societal Impact:**

No section is provided, the paper could benefit from discussing this.

**Main Review:**

Strengths: This paper works on a challenging task that is of interest to many in the community, and it has competitive results. Additionally, many resaerchers in both language and vision are interested in combining syntax with deep-learning based approaches, so this learning paradigm could be interesting to them as well.

In addition to strong results (both human as well as automatic eval), an ablation study is shown. The ablations seem to suggest that using a tree structure helps, particularly when RL is optimized. Qualitative examples are also discussed.

Overall, I would be willing to raise my score and vote to accept this paper if the weaknesses below are addressed.

Weaknesses: The main concern to this reviewer concerns the overall premise of the paper. The argument expressed in L29-L30 is that graph-structure on the text allows the model to resolve long-term dependencies. However, the language in these datasets is pretty short -- 1 to 2 sentences. Thus, it seems very likely to this reviewer that there is something else going on for why in the ablation study, the tree model outperforms the baselines. I would like to be surprised, however, this paper would benefit from convincing evidence for why the tree model is better, along with helpful explanation about what the tree-structured losses/model achieves exactly.

The ablations are helpful, but one ablation that the paper could benefit from is disentangling the objective from the model in some way. In other words, if the model was trained to jointly generate trees, as well as sentences, and at test time generated only one of those, that might help provide evidence for why a tree structure is better. An alternate explanation for the ablation results might be that the losses proposed add more regularization to the model, even if they are not actually helpful for the end task.

Section 4.6 claims that the proposed tree structure is better at generating complex sentences, however, this seems a bit unclear from Table 4 alone. Though the model seems to degrade less when compared with references that are longer, to this reviewer this is not by itself indicative of generating longer sentences, but rather being more similar to other longer sentences. One way the argument might be made stronger is if you evaluated the sentence length or complexity of generated sentences in some way, comparing "w/o tree" with the proposed model.


**Time Spent Reviewing:**

1

---

> ### Author Response · Authors · 2021-08-10
> **Response to Reviewer NzTM**
>
> We appreciate reviewer NzTM for the positive comments of "interesting learning paradigm" and "competitive results" for our paper. We address the questions below and hope the rating can be improved.
>
> **Q1:** *The main concern to this reviewer concerns the overall premise of the paper. The argument expressed in L29-L30 is that graph-structure on the text allows the model to resolve long-term dependencies. However, the language in these datasets is pretty short -- 1 to 2 sentences. Thus, it seems very likely to this reviewer that there is something else going on for why in the ablation study, the tree model outperforms the baselines. I would like to be surprised, however, this paper would benefit from convincing evidence for why the tree model is better, along with helpful explanation about what the tree-structured losses/model achieves exactly.*
>
> **A1:**
> We appreciate your constructive suggestion. In comparison with the baselines, our tree model uses the same visual features and the same encoder architecture as them. So the reason why the tree model outperforms the baselines is the tree-structured decoder that better captures the syntactic structure of the sentence. In other words, the quality of the output sentence depends on the generated tree.
>
> To further show the contribution of the tree structure, we provide the tree structure evaluations using a metric based on the edit distance between dependency trees [McCaffery et al. 2016]. Please refer to the answer A3 to the question Q3 of reviewer xmz3 for more details.
>
> What's more, the dependency tree is generated in top-down order and our tree-structured decoder is able to accurately determine the key semantics of the sentence by generating the most important words (verbs and nouns) in the first few steps of the decoding process.
> In contrast, the sequence generation model follows the left-to-right order and alternately generates the important words and other words (adjectives, adverbs, etc.).
> To validate this hypothesis, we remove all other words in ground-truth sentences and generated sentences, and evaluate the quality of the generated nouns and verbs on Charades Captions.
> From the results shown below, we observe that the tree-structured decoder generates the nouns and verbs more accurately, thus improving the captioning performance.
>
> | model            | B@4       | M     | R     | C     |
> | ---------------- | ------    | ----- | ----- | ----- |
> | w/o tree         | 7.32      | 16.24 | 31.78 | 24.94 |
> | Ours             | 8.02      | 16.84 | 33.22 | 26.38 |
>
> [McCaffery et al. 2016] DTED: evaluation of machine translation structure using dependency parsing and tree edit distance. In Proceedings of the First Conference on Machine Translation
>
> **Q2:** *The ablations are helpful, but one ablation that the paper could benefit from is disentangling the objective from the model in some way. In other words, if the model was trained to jointly generate trees, as well as sentences, and at test time generated only one of those, that might help provide evidence for why a tree structure is better. An alternate explanation for the ablation results might be that the losses proposed add more regularization to the model, even if they are not actually helpful for the end task.*
>
> **A2:** Thank you for your constructive suggestion.
> We train the proposed model to generate trees and sequences at the same time, and the results of generating trees and sequences using this model (denoted as Ours (joint) ) are shown below.
> We observe that by jointly training the model to generate trees and sequences, the performance of generating sequences slightly improves, which indicates that the losses for constructing the trees is also helpful for generating sequences.
>
> | Method       | generation | B@4       | M    | R    | C    |
> | ------------ | ---------- | ------    | ---- | ---- | ---- |
> | w/o tree     | sequence   | 17.7      | 18.7 | 40.2 | 23.5 |
> | Ours         | tree       | 18.9      | 19.8 | 41.7 | 24.2 |
> | Ours (joint) | sequence   | 17.9      | 18.7 | 40.6 | 23.7 |
> | Ours (joint) | tree       | 18.8      | 19.8 | 41.7 | 24.3 |
>
>
> **Q3:** *Section 4.6 claims that the proposed tree structure is better at generating complex sentences, however, this seems a bit unclear from Table 4 alone. Though the model seems to degrade less when compared with references that are longer, to this reviewer this is not by itself indicative of generating longer sentences, but rather being more similar to other longer sentences. One way the argument might be made stronger is if you evaluated the sentence length or complexity of generated sentences in some way, comparing "w/o tree" with the proposed model.*
>
> **A3:** We calculated the average lengths of the generated sentences and the ground-truth sentences on MSR-VTT and Charades and the results are shown below. We observe that the sentences generated on the complex subsets are indeed longer than the sentences generated on simple subsets.
>
> | Subset             | Ground-truth | w/o tree | Ours   |
> | :-- | :-- | :-- | :-- |
> | charades (simple)  | 17.780       | 17.591   | 21.370 |
> | charades (complex) | 25.071       | 18.853   | 21.894 |
> | MSR-VTT (simple)   | 7.630        | 7.586    | 6.959  |
> | MSR-VTT (complex)  | 12.746       | 8.668    | 7.890  |

---

> > ### Comment · Reviewer_NzTM · 2021-08-27
> > **Thanks for the response! Increasing my score from 5->7**
> >
> > Thanks for the helpful response! I read it, as well as the other reviews (and responses to them).
> >
> > My main concern was that the premise of the paper is sketchy (we need special handling for long-term text dependencies in video captioning, even though e.g. unstructured LMs can generate ~hundreds to thousands of tokens). I think that concern was partially addressed through experimentation. The experiments seem to suggest that the model helps in a way that goes beyond regularization, however, the gap of ~1pt B@4, and ~0.6 pts CIDEr is possibly not super significant. My intuition here is thus unchanged for now, but I think this paper could prompt some more exploration in this area and thus could be a good addition to NeurIPs, so I'll increase my score to 7 (slight acceptance).
> >
> > The model seems to outperform SOTA models, and does well on image captioning too, which appears to me like it answers the key concerns of other reviewers.

---

### Official Review · Reviewer_xmz3 · 2021-07-16

**Rating:** 6
**Confidence:** 4

**Summary:**

The paper addresses video captioning by generating captions in a tree structure. It stepwise constructs the multi-modal dependency tree and then traverses the constructed tree to obtain the captions. The supervision is performed on generated captions as well as the semantic consistency reward of the generated sub-trees. Experiments are conducted on several video captioning datasets.

**Limitations And Societal Impact:**

Authors do not discuss the limitation and potential negative societal impact of their work.

**Main Review:**

Strengths

+ The idea of tree-structured caption generation is novel and reasonable for video captioning, which could explicitly model the hierarchical structure in the sentence and capture the long-range dependency between words.

+ The paper is well-organized and clearly written, which can be followed easily.

+ Experiments are conducted on several video captioning datasets, and the ablation study shows the effectiveness of the tree-structure generation and other modules.

+ For a fair comparison with state-of-the-art methods, authors evaluate their model using different visual features.

Weaknesses

+ The authors do not compare their methods with some state-of-the-art methods, such as [1], ORG-TRL[2], PMI-CAP[3], RMN(H+L)[34], SBAT[4], etc. And those methods outperform the proposed model. Authors should include them and discuss the difference and reason.

+ Authors do not provide the intermediate results of tree-structure caption generation, and those intermediate results are essential for readers to understand how the hierarchical structure contributes to captioning.

+ The method could also be adapted to image captioning. The authors are suggested to provide the results for image captioning to demonstrate the effectiveness of the proposed method further.

[1] Hou et al., Joint Syntax Representation Learning and Visual Cue Translation for Video Captioning. ICCV2019

[2] Zhang et al., Object Relational Graph with Teacher-Recommended Learning for Video Captioning. CVPR2020

[3] Chen et al., Learning Modality Interaction for Temporal Sentence Localization and Event Captioning in Videos. ECCV2020

[4] Jin et al., SBAT: Video Captioning with Sparse Boundary-Aware Transformer. IJCAI2020


**Time Spent Reviewing:**

7 hours

---

> ### Author Response · Authors · 2021-08-10
> **Response to Reviewer xmz3**
>
> We appreciate reviewer xmz3 for the positive comments on novelty, presentation and experiments of our paper. We address the questions below and hope the rating can be improved.
>
> **Q1:** *The authors do not compare their methods with some state-of-the-art methods, such as [1], ORG-TRL[2], PMI-CAP[3], RMN(H+L)[34], SBAT[4], etc. And those methods outperform the proposed model. Authors should include them and discuss the difference and reason.*
>
> **A1:** [1] has been compared in the paper (i.e., Joint[11] in Table 3).
> We conduct additional experiments to compare our method with ORG-TRL, PMI-CAP, RMN(H+L), and SBAT.
> For fair comparison with them, we use the same features as them and the results are shown below.
> We have the following observations: (1) on the MSR-VTT dataset, which is more complex, our method achieves comparable performance with the SOTA methods for most evaluation metrics; (2) on the MSVD dataset, RMN(H+L), ORG-TRL and PMI-CAP outperform our method, probably due to that RMN(H+L) and ORG-TRL utilize additional object features extracted by Faster R-CNN (denoted as Obj in the table), and PMI-CAP uses more superior interaction technique between appearance and motion features while our method simply concatenates these features. More detailed analysis and discussions of the results will be added in the revised version.
>
>
> |                   |           | MSVD |      |      |      | MSR-VTT |      |      |      |
> | ----------------- | --------- | ---- | ---- | ---- | ---- | ------- | ---- | ---- | ---- |
> | Feature           | Model     | B@4  | R    | M    | C    | B@4     | R    | M    | C    |
> | IRV2+I3D(RGB)+Obj | RMN (H+L) | 54.6 | 73.4 | 36.5 | 94.4 | 42.5    | 61.6 | 28.4 | 49.6 |
> | IRV2+I3D(RGB)     | SBAT      | 53.1 | 72.3 | 35.3 | 89.5 | 42.9    | 61.5 | 28.9 | 51.6 |
> | IRV2+I3D(RGB)     | ours      | 51.8 | 72.8 | 36.7 | 92.5 | 42.6    | 61.8 | 29.3 | 49.7 |
> | IRV2+C3D+Obj      | ORG-TRL   | 54.3 | 73.9 | 36.4 | 95.2 | 43.6    | 62.1 | 28.8 | 50.9 |
> | IRV2+C3D          | PMI-CAP   | 54.7 | \-   | 36.4 | 95.2 | 42.2    | \-   | 28.8 | 49.5 |
> | IRV2+C3D          | ours      | 52.0 | 72.9 | 36.5 | 92.6 | 42.7    | 61.9 | 29.4 | 50.1 |
>
>
> **Q2:** *Authors do not provide the intermediate results of tree-structure caption generation, and those intermediate results are essential for readers to understand how the hierarchical structure contributes to captioning.*
>
> **A2:**
> Thank you for your valuable suggestion. We provide the evaluations of the dependency structure as the intermediate results since the quality of the output sentence depends on the generated dependency tree.
> An evaluation metric, DTED [McCaffery et al. 2016] (lower is better) that calculates the weighted and normalized tree edit distance between the dependency trees of the generated sentences and ground-truth sentences.
> For "w/o tree", we use the dependency parser in SpaCy toolkit to convert the sentences to dependency trees for evaluation.
> The results of "w/o tree" and "Ours" are shown below, where the syntactic structure of the sentences generated by "Ours" is more accurate, leading to better captioning performance.
> In addition, we have provided some qualitative results of the generated tree structure in Figure 1 and Figure 2 in the supplementary material.
>
> Method      | DTED ($\downarrow$)  |
> --:         | --:   |
> w/o tree    | 12.02 |
> Ours        | 10.73 |
>
> [McCaffery et al. 2016] DTED: evaluation of machine translation structure using dependency parsing and tree edit distance. In Proceedings of the First Conference on Machine Translation.
>
> **Q3:** *The method could also be adapted to image captioning. The authors are suggested to provide the results for image captioning to demonstrate the effectiveness of the proposed method further.*
>
> **A3:** Thank you for your constructive suggestion. We conduct additional image captioning experiments on MSCOCO dataset using ResNet-101 features and the results are shown below.
>
> | Method   | B@4   | M     | R     | C      |
> | -------- | ----- | ----- | ----- | ------ |
> | w/o tree | 36.26 | 27.21 | 56.38 | 119.52 |
> | Ours     | 37.03 | 27.87 | 56.42 | 120.30 |
>
> We also conducted preliminary experiments on the Stanford Image-paragraph dataset [Krishna et al. 2017], and the results are shown below.
>
> | Model    | B@4  | M     | C     |
> | -------- | ---- | ----- | ----- |
> | w/o tree | 7.94 | 16.56 | 19.75 |
> | Ours     | 9.96 | 18.95 | 21.30 |
>
> On both image captioning datasets, we observe that: (1) introducing the tree structure improves the performance of image captioning, and (2) our method is better at generating long captions, such as the paragraphs in the Stanford Image-paragraph dataset that consists of 5 to 7 sentences on average.
>
> [Krishna et al. 2017] Krishna et al. Dense-Captioning Events in Videos. In ICCV 2017.

---

> > ### Comment · Reviewer_xmz3 · 2021-09-11
> > **After Rebuttal**
> >
> > Thank the authors for their effort and time during the rebuttal. After reading other reviews and the authors’ replies to all the reviews, I decided to keep my initial rating as 6, i.e., borderline accept.
> >
> > The authors tried to answer my questions in the rebuttal, but their answers to my Q1(performance compared to some SOTAs) and Q3(generalization to image captioning) are not convincing enough. However, considering the novelty and insight are inspiring enough, I would keep my rating of borderline accept.

---

> > > ### Comment · Area_Chair_gVQN · 2021-09-11
> > > **Why not convincing enough?**
> > >
> > > Hi, thanks for response, Reviewer xmz3!
> > >
> > > Can you please be specific why the comparison to SOTA and image captioning is "not convincing enough"?
> > > Is it the performance difference? is it the comparisons done? what would have convinced you?...
> > >
> > > Your Meta Reviewer

---

### Official Review · Reviewer_c8JF · 2021-07-16

**Rating:** 5
**Confidence:** 3

**Summary:**

This paper propose a video captioning method that generates a sentence by first constructing a multi-modal dependency tree and then traversing the constructed tree.Both the visual and textual representation features are encoded into each tree node to construct a multi-modal dependency trees for language generation of videos. The syntactic structure and semantic relationship in the sentence are represented by the tree topology.

**Ethics Review Area:**

["I don’t know"]

**Limitations And Societal Impact:**

1.The motivation to investigate a graph structured model is to capture the global dependency structure in the sentence which different from existing sequence models that tend to focus on the dependency between each word and its close preceding words. However,the encoder and decoder is based on Transformer，which can draw global dependencies in sentence. Therefore, I am a bit confused about the complex approach of this article.

2.Table3 does not use the C3D feature to do the experiment, I think this paper should compare the result of C3D features in MSR-VTT with [1].On the MSR-VTT dataset, the performance of the method has not improved much on CIDEr, within 0.3%.

3.In the ablation experiment, the performance without reinforcement learning dropped lower than without dependency tree.The two tables do not list the cases where dependency tree and RL are not used.

4.Constructing the generation tree, constructing the ground-truth dependency trees, calculating loss and reward all consume many computing resources. Is the time cost of the training process large?

[1] Zhang at al.Object Relational Graph with Teacher-Recommended Learning for Video Captioning.CVPR'2020
Typos:e.g.The last two rows of table3 are wrongly blackened on the METEOR on dataset MSR-VTT.

**Main Review:**

1.The amount of engineering for this paper is large.This paper has a large amount of experimentation，the author has done multiple experiments on both complex and simple datasets.There are many hyper-parameters, which I personally think is difficult to reproduce. Will the code be provided later?

2.This paper develop a novel tree-structured reinforcement learning algorithm that optimizes the captioning model using a node-level reward, which facilitates learning the topology of the dependency trees and alleviates the reward ambiguity problem.

**Time Spent Reviewing:**

3

---

> ### Author Response · Authors · 2021-08-10
> **Response to Reviewer c8JF**
>
> We appreciate the positive comments and valuable suggestions of Reviewer c8JF. We address the questions below and hope the rating can be improved.
>
> **Q1:** *There are many hyper-parameters, which I personally think is difficult to reproduce. Will the code be provided later?*
>
> **A1:** We will release the code, data and models when the paper is published.
>
> **Q2:** *The motivation to investigate a graph structured model is to capture the global dependency structure in the sentence which different from existing sequence models that tend to focus on the dependency between each word and its close preceding words. However,the encoder and decoder is based on Transformer, which can draw global dependencies in sentence. Therefore, I am a bit confused about the complex approach of this article."*
>
> **A2:**
> The proposed dependency tree not only captures the global dependencies between words but also models the hierarchical structure of the sentence.
> Though the original Transformer model also considers the dependencies in the sentence, it treats the sentence as a word sequence and is unaware of the syntactic structure.
>
> **Q3:** *Table3 does not use the C3D feature to do the experiment, I think this paper should compare the result of C3D features in MSR-VTT with [1]. On the MSR-VTT dataset, the performance of the method has not improved much on CIDEr, within 0.3%.*
>
> *[1] Zhang at al. Object Relational Graph with Teacher-Recommended Learning for Video Captioning. CVPR'2020*
>
> **A3:** Thank you for your constructive suggestion. The experiment results using the same feature (InceptionResNetv2 + C3D) as [1] on MSR-VTT are shown below.
> We observe that our method reaches comparable performance with ORG-TRL [1] when using C3D features.
> Please note that the encoder of ORG-TRL uses additional object features extracted by Faster RCNN (denoted as Obj in the table), which contributes to its performance.
>
> | Feature      | Model   | B@4     | R    | M    | C    |
> | ------------ | ------- | ------- | ---- | ---- | ---- |
> | IRV2+C3D+Obj | ORG-TRL | 43.6    | 62.1 | 28.8 | 50.9 |
> | IRV2+C3D     | ours    | 42.9    | 62.0 | 29.3 | 50.3 |
>
> **Q4:** *In the ablation experiment, the performance without reinforcement learning dropped lower than without dependency tree.The two tables do not list the cases where dependency tree and RL are not used.*
>
> **A4:** We conduct additional ablation experiments where both the dependency tree and RL are removed, denoted as "w/o tree, RL". The results on ActivityNet Captions and Charades Captions are shown below. The "w/o tree, RL" is worse than "w/o tree" and "w/o RL" on all the metrics, which indicates that both the tree structure and reinforcement learning contribute to the captioning performance.
>
> Model           | B@4   | M     |C
> :-- | --: | --: | --:
> w/o tree, RL    | 1.38  | 10.15 | 37.04
> w/o tree        | 1.47  | 12.40 | 42.30
> w/o RL          | 1.40  | 11.23 | 38.90
> Ours            | 1.75  | 13.24 | 44.13
>
> [results on ActivityNet Captions]
>
> Model           | B@4   | M     |C
> :-- | --: | --: | --:
> w/o tree, RL    | 1.38  | 10.15 | 37.04
> w/o tree        | 1.47  | 12.40 | 42.30
> w/o RL          | 1.40  | 11.23 | 38.90
> Ours            | 1.75  | 13.24 | 44.13
>
> [results on Charades Captions]
>
>
> **Q5:** *Constructing the generation tree, constructing the ground-truth dependency trees, calculating loss and reward all consume many computing resources. Is the time cost of the training process large?*
>
> **A5:** The time cost of the training process is not large.
> The time complexity of constructing the generation tree is $O(n)$, where n is the number of the tree nodes (i.e., the number of the words in the sentence).
> The construction of the ground-truth dependency trees is done before training and costs no extra training time.
> The times for calculating the cross-entropy loss and the reward of the tree-structured model are also similar to the times when training a sequence model.
> In practice, using a GeForce RTX 2070 GPU, training 10 epochs on the Charades Captions dataset with cross-entropy loss and reinforcement learning takes about 25 minutes and 80 minutes, respectively.
> The entire training process on Charades Captions takes about 4 hours.

---

### Decision · Program_Chairs · 2021-09-27

**Decision:**

Accept (Poster)

**Comment:**

All reviewers who took part in the post rebuttal discussion recommend or lean to accept the paper. In my opinion, the concerns of reviewer c8JF have also been addressed.

The paper contributes an interesting model of using dependency trees for video captioning, showing solid performance compared to SOTA, and clear experimental ablations. Additional results and ablations, including on image captioning in the author response strengthen the paper further.

I recommend accepting the paper under the expectation that the authors address the concerns of the reviewers as done in the author response, including, but not limited to the following
1) include additional results and ablations (if they don't fit include them in supplement).
2) promptly release code, data and models when the paper is published.
3) include clarifications and typos / errors (e.g. bolding in Table 3)